# Zooplankton-phytoplankton biomass and diversity relationships in the Great Lakes

Katya E. Kovalenko[1]*, Euan D. Reavie[1], Stephanie Figary[2], Lars G. Rudstam[2], James M. Watkins[2], Anne Scofield[3], Christopher T. Filstrup[1]

1 Natural Resources Research Institute, University of Minnesota, Duluth, MN, United States of America, 2 Department of Natural Resources and Cornell Biological Field Station, Cornell University, Ithaca, NY, United States of America, 3 U.S. EPA Great Lakes National Program Office, Chicago, IL, United States of America

* philarctus@gmail.com

**Data Availability Statement:** All relevant data are available within the manuscript and its Supporting Information files.

## Abstract

Quantifying the relationship between phytoplankton and zooplankton may offer insight into zooplankton sensitivity to shifting phytoplankton assemblages and the potential impacts of producer-consumer decoupling on the rest of the food web. We analyzed 18 years (2001–2018) of paired phytoplankton and zooplankton samples collected as part of the United States Environmental Protection Agency (U.S. EPA) Great Lakes Biology Monitoring Program to examine both the long-term and seasonal relationships between zooplankton and phytoplankton across all five Laurentian Great Lakes. We also analyzed effects of phytoplankton diversity on zooplankton biomass, diversity, and predator-prey (zooplanktivore/grazer) ratios. Across the Great Lakes, there was a weak positive correlation between total algal biovolume and zooplankton biomass in both spring and summer. The relationship was weaker and not consistently positive within individual lakes. These trends were consistent over time, providing no evidence of increasing decoupling over the study period. Zooplankton biomass was weakly negatively correlated with algal diversity across lakes, whereas zooplankton diversity was unaffected. These relationships did not change when we considered only the edible phytoplankton fraction, possibly due to the high correlation between total and edible phytoplankton biovolume in most of these lakes. Lack of strong coupling between these producer and consumer assemblages may be related to lagging responses by the consumers, top-down effects from higher-level consumers, or other confounding factors. These results underscore the difficulty in predicting higher trophic level responses, including zooplankton, from changes in phytoplankton assemblages.

## Introduction

With a few rare exceptions, aquatic ecosystems in the Anthropocene have experienced changes in temperature and nutrient concentrations, which can lead to shifts in phytoplankton assemblages [1–3]. In many cases, these compositional changes can alter the seasonal timing and amplitude of primary productivity [4, 5] and functional attributes of phytoplankton [6, 7].

**Funding:** These data were collected as part of the U.S. Environmental Protection Agency's (EPA's) Great Lakes Biology Monitoring Program. Thus, the study design for sample collection and taxonomic analysis to evaluate phytoplankton and zooplankton communities was determined by the EPA, and followed methods specified by the standard operating procedures associated with this program. The funder did not determine the data analysis method, decision to publish, or assist with preparation of the manuscript beyond the scope of the contributing author affiliated with EPA.

**Competing interests:** The authors have declared that no competing interests exist.

Changes in phytoplankton assemblage composition and dynamics can lead to decoupling of primary producers and consumers, which may destabilize planktonic food webs with cascading effects on tertiary consumers [8–10].

Theory predicts and observational studies have shown that greater phytoplankton diversity is linked to increased phytoplankton resource use efficiency (horizontal diversity effects within trophic levels, [11]) and to increased zooplankton growth rate, diversity, and abundance (vertical diversity effects across trophic levels [12]). Phytoplankton diversity can also directly influence consumers via biochemical diversity in food resources, which should increase zooplankton diversity [13], and these diversity effects may produce direct and indirect feedbacks to buffer primary consumer populations and entire food webs from abrupt shifts in their resource base. Because phytoplankton diversity can decrease variability in zooplankton productivity [12], greater algal diversity may support more zooplankton predators and therefore greater predator-prey ratios within the zooplankton community. However, diversity effects are not consistent across systems [14] and different measures of phytoplankton diversity can have opposing influences on horizontal and vertical diversity effects [15]. For example, communities dominated by cyanobacteria may have larger proportions of inedible taxa [16], which might limit zooplankton biomass [17, 18] or have no impact [19]. Predator-prey biomass ratios can respond to environmental stressors when predators take longer to recover from perturbations, e.g., in isolated environments [20]; however, other studies show remarkable consistency in predator-prey ratios across a wide range of taxa and systems [21, 22].

The structure of large lake food webs is less understood than that of smaller lakes [23, 24], and previous vertical diversity studies have largely focused on smaller ecosystems. In the Laurentian Great Lakes, several attributes of phytoplankton assemblages, including total biovolume, cell densities, average cell sizes, and species composition, have fluctuated considerably in the last few decades, with likely causes being changes in nutrient availability, invasive species, and climate change [5, 25–27]. Decreasing algal cell sizes in particular [27] could have repercussions for the entire aquatic food web, consistent with a climate change signal linked to decreasing organism sizes at community, species, and population levels across a range of ecosystems [28]. In the Great Lakes, zooplankton shifted to greater dominance by calanoid copepods, particularly *Limnocalanus macrurus* [29], abundances of the predatory invasive cladoceran *Bythotrephes* increased in some lakes [30], causing declines in some species [31–33] and changes vertical distribution in others due to migration to greater depths as an antipredatory response to *Bythotrephes* [31]. With a wealth of long-term historical data, there have been multiple detailed analyses of trends in specific assemblages [25, 34, 35] and concurrent trends [36, 37]; however, the degree of zooplankton and phytoplankton coupling, vertical diversity effects, and detailed associations between specific groups of taxa are less well understood.

Ideally, investigations of the relationships between primary producers and consumers should use high-resolution productivity data and information on feeding selectivity [38]. However, long-term high-resolution *in situ* productivity data are relatively sparse and often limited to smaller geographic areas (e.g., [39]), and landscape-scale analyses often rely on standing biomass. Controlled studies of feeding selectivity, usually conducted in laboratory settings, are similarly difficult to extrapolate to diverse and dynamic natural settings. We used nearly 20 years of paired zooplankton and phytoplankton data from the U.S. EPA Great Lakes Biology Monitoring Program to examine ecological associations, long-term and seasonal dynamics of zooplankton-phytoplankton coupling, and effects of phytoplankton diversity on zooplankton biomass and diversity. We predicted that there would be a positive correlation between algal biovolume and zooplankton biomass, and that the slope of this relationship would decrease over time because of increasing decoupling of the two trophic levels associated with changes in

phytoplankton assemblages. We also tested relationships between algal diversity and total zooplankton biomass, zooplankton diversity, and zooplanktivore-grazer ratios, and explored group-level associations between the major types of zooplankton and algae.

## Materials and methods

We used data collected as part of the U.S. Environmental Protection Agency (EPA) Great Lakes Biology and Water Quality Monitoring Programs in the pelagic Laurentian Great Lakes of North America, focusing on years which had matching phytoplankton and zooplankton data (2001–2018). Samples are collected twice per year in the spring (usually April) and summer (usually August) from 72 sites across the five Great Lakes: Lakes Erie, Ontario, Huron, Michigan, and Superior (S1 Table). For phytoplankton, equal volumes of water were collected by a rosette sampler from multiple depths (0, 5, 10, 20 m) at each station representing the upper 20 m of the isothermal water column in the spring or the epilimnion in the summer [25]. Four spring samples from individual depths were composited to form an integrated sample; in summer, a minimum of two and maximum of four depths (typically 0, 5, 10 m, and lower epilimnion, but fewer taken when the mixed layer is shallow) were composited to form a representative sample from the epilimnion [40]. Samples were preserved with Lugol's iodine solution and analyzed as described in U.S. EPA Great Lakes National Program Office (GLNPO) standard operating procedure [41]. Briefly, we used the Utermöhl method [42] for soft-bodied algal identification. Subsamples were processed for detailed diatom assessment by acid digestion, slide-mounting and high-resolution microscopy. Algal specimens were also measured to allow for biovolume calculations [43].

Phytoplankton taxa were characterized as edible or inedible based on a combination of entity shape and nutritional quality. Characterization of edibility in freshwater phytoplankton has been considered previously [44], and we followed similar methods. We assumed that cyanobacteria are less desirable food organisms due to their poor nutritional quality [45]. Further, we considered a prevailing size and shape of entities (as single cells, filaments, globular colonies) greater than 50 μm to be inedible. Therefore, algae such as filamentous diatoms are considered problematic as food for zooplankton despite their high nutritional value. We acknowledge that previously published assumptions around edibility are overly simplistic, and that edibility of a given phytoplankton taxon is likely grazer-specific. For instance, some larger zooplankton taxa may be equipped to disaggregate large, filamentous diatoms into edible sizes, as noted in a limited set of species-specific studies from marine systems (e.g., [46]). Such nuances should be considered in the future, but we treat our analyses as a first attempt to evaluate this phenomenon in the Great Lakes. Using these edibility criteria, we filtered out all phytoplankton taxa with low nutritional and low shape edibility (S2 Table), and recalculated biovolume of remaining phytoplankton at each site.

Crustacean zooplankton and rotifers were collected by vertical tows taken across the same depth range, at the same time and stations as the phytoplankton data. All samples were collected according to U.S. EPA GLNPO standard operating procedure LG402 [47] and analyzed following LG403 [48]. Samples used here were collected using a 63 μm mesh net towed from 20 m or 1 m above the bottom, whichever was shallower, to the surface, at a rate of 0.5 m/s. As with phytoplankton, zooplankton sample collection for this program occurs 24 hours a day, and some stations are sampled during the day and some at night. Zooplankton samples from 20 m were not available for 2007 (both seasons) and for the spring season 2008–2011, and fewer stations had matching data for the two assemblages earlier in the time series. Plankton were narcotized with soda water and preserved with sucrose formalin. Separate counts with different subsampling approaches were done for crustaceans and microzooplankton (rotifers,

nauplii) and data combined to densities (numbers/ m$^3$). A minimum of 400 individuals for each of the two counts were identified to the smallest practical taxonomic unit (mostly species) and up to 20 individuals in each taxonomic unit were measured for length in mm using a computerized drawing tablet [48]. Dreissenid veligers were not included in the total biomass calculations because they have not been measured consistently across the years (sensitivity analysis demonstrates that < 2% of the site-years are affected by this bias). Dry weight individual biomass (μg) was calculated from taxa-specific length-weight regressions available in the standard operating procedures [29, 48]. Some rotifer equations used width measurements.

### Statistical analyses

We used simple linear models to test for correlations between phytoplankton biovolume and zooplankton biomass, and correlations between phytoplankton and zooplankton diversity (Shannon H). All biovolume analyses were repeated with total and edible phytoplankton biovolume. In addition, we tested the relationships between zooplankton excluding predatory cladoceran (*Bythotrephes*, *Cercopagis*, *Leptodora* and *Polyphemus*) and *Limnocalanus* biomass, and edible algal biovolume and diversity, although *Limnocalanus* varies in its degree of zooplanktivory across the Great Lakes [49]. Data distribution was checked using *qqnorm* function in R and log$_{10}$-transformation was applied to reduce skewness when warranted (biovolume and biomass data). Additionally, Spearman cross-correlation analyses were used to visualize the relationships between key groups of phytoplankton and zooplankton (S2 Table). Zooplankton predator ratios were calculated using the sum of predatory cladocerans (*Bythotrephes*, *Cercopagis*, *Leptodora* and *Polyphemus*) and *Limnocalanus* biomass relative to other zooplankton. Generalized additive models were fitted to visualize zooplankton predator-prey relationships with algal community metrics; model parameters were set to default as passed on to geom_smooth function in ggplot2. Time of day analyses were used to understand the relative importance of sampling time on zooplankton biomass-edible algal biovolume correlations within lakes. All analyses were done in R [50].

## Results

Across the 20 years of data and all of the lakes, there was a weak positive correlation between total algal biovolume and zooplankton biomass (P < 0.0001, R$^2$ = 0.19). This Great Lakes-wide correlation was season-dependent, with the overall trend driven primarily by the summer (P < 0.0001, R$^2$ = 0.15, vs. spring R$^2$ = 0.06, Fig 1). The relationship was also scale-dependent and varied across individual lakes, with a positive relationship in Lake Erie in both seasons and in Lake Huron in the spring, but a lack of significant correlations in lakes Ontario, Superior and Michigan in either season (Fig 1). The slopes of biomass-biovolume relationship (evidence of decoupling) did not change uniformly with time (P > 0.05, Fig 2).

Total zooplankton biomass was very weakly negatively correlated with phytoplankton Shannon diversity (P = 0.001, R$^2$ < 0.01) and this relationship was similarly weak in both spring and summer across all lakes (Fig 3). This weak negative effect was driven largely by Lake Erie, which spanned the longest gradient of both biomass and diversity, and was less pronounced in other lakes. Zooplankton diversity was likewise very weakly correlated with phytoplankton diversity (R$^2$ < 0.02, S1 Fig). Most of the biomass of different zooplankton groups was unrelated or weakly negatively related to overall algal richness and diversity, with the exception of a stronger positive relationship for *Limnocalanus* (R$^2$ = 0.15, Fig 4). The majority of zooplankton groups had closer associations with other zooplankton groups (e.g., predatory cladoceran and rotifers), followed by biovolumes of Cyanophyta, Chlorophyta, and total algal biovolume (Fig 4). Some variation in zooplankton predator-prey ratios was explained by algal

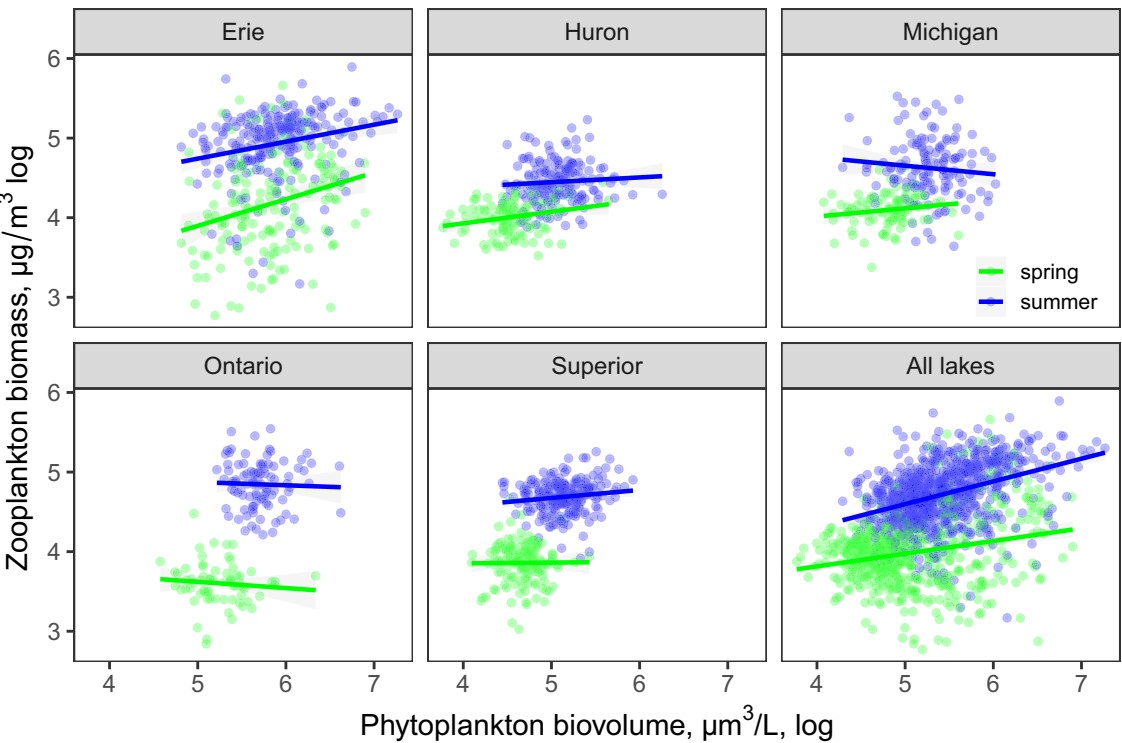

**Fig 1. Phytoplankton biovolume and zooplankton dry biomass correlations.** Data are presented for all lakes and for individual Laurentian Great Lakes by season.

diversity (P < 0.0001), whereas algal richness and biovolume did not have a strong linear effect (S2 Fig).

Edible algal biovolume was closely correlated with the overall algal biovolume (across lakes $R^2$ = 0.78, P < 0.0001 in each lake), with the largest discrepancy observed for Lake Erie (Fig 5), where cyanobacteria are abundant in the summer. Our edibility criteria excluded algae with low nutritional value as well as those with difficult to manipulate shapes; we did not consider the two types of edibility filters separately, because even in the extreme scenario, there was a close relationship with total algal biovolume. Because of this relatively high correlation, most of the zooplankton-phytoplankton relationships were not greatly affected when considering only edible phytoplankton biovolume (S3 and S4 Figs). Results of analyses excluding predatory cladocerans and *Limnocalanus* detected similarly weak trends to those for total zooplankton biomass (S5 and S6 Figs). Examining zooplankton-phytoplankton relationship by the time of sampling demonstrated relatively minor effects of time of day on the shape of the biovolume-biomass relationship in individual lakes (S7 Fig). The relationship between total and edible biovolume did not exhibit directional changes over time (S8 Fig).

## Discussion

There was a statistically significant but weak correlation between phytoplankton biovolume and zooplankton biomass across this long-term, large-scale dataset; however, it only held across the entire basin, and not individual lakes, and only in the summer. The weak correlation between phytoplankton biovolume and zooplankton biomass on a lake by lake basis could result from a lag in the response of zooplankton consumers to algal changes or variable top-

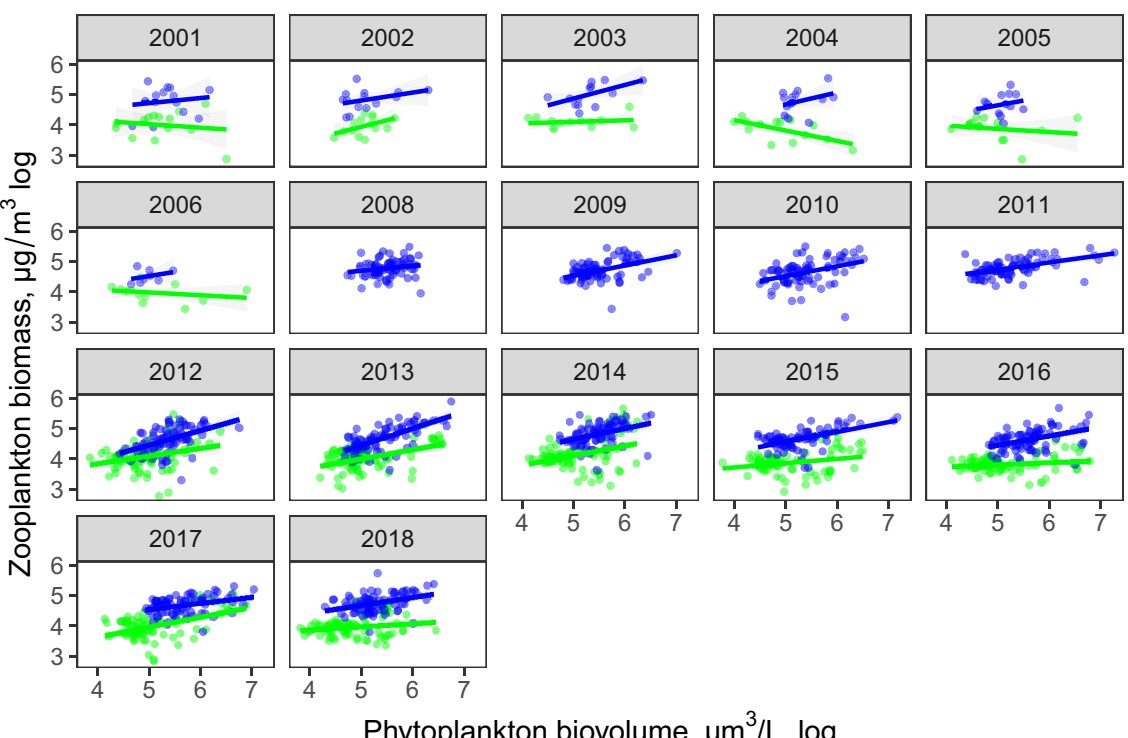

**Fig 2. Temporal dynamics of the phytoplankton-zooplankton relationship.** Fewer stations had matching data for the two assemblages earlier in the time series and spring data was unavailable for zooplankton between 2008–2011 (see S1 Table for complete summary of stations sampled by year).

down forcing on zooplankton across the lakes. If a lag in consumer response is present, we would expect the relationship to be stronger in the summer, which was generally the case, even though the correlation was still very poor in terms of predictive power and not statistically significant for most individual lakes. It is not surprising that the large trophic gradient of these lakes, from oligotrophic to meso-eutrophic, was also reflected by the gradient in zooplankton biomass and phytoplankton biovolume across the entire basin. Similarly, in other lakes the coupling between phytoplankton biomass and zooplankton biomass was limited beyond a certain productivity level [51, 52 for Lakes Balaton and Lake Constance].

The slope and strength of the relationship between phytoplankton and zooplankton did not vary significantly with time, despite considerable shifts in algal and zooplankton community composition and productivity [5, 25, 29], providing little additional evidence for a disruption in coupling of producers and consumers. The match/mismatch hypothesis focuses on the consequences of inter-specific differences in response to climate change leading to potentially non-linear responses in the patterns of synchrony [9]. Such decoupling has already been observed in other systems as a result of a mismatch between trophic levels responding primarily to photoperiod vs. those responding to temperature [53]. In temperate lakes, the timing of thermal stratification affects the spring diatom blooms which are increasingly mismatched with keystone consumer dynamics [54]. In the Great Lakes, decreasing diatom cell sizes due to accelerated loss of larger individuals during summer stratification [27], for example, could make consumers rely on less energetically optimal smaller-sized algae. Longer ice-free periods in Lake Superior have resulted in longer stratification and increased primary production [5] and could lead to a timing mismatch between the peak of the spring bloom and zooplankton

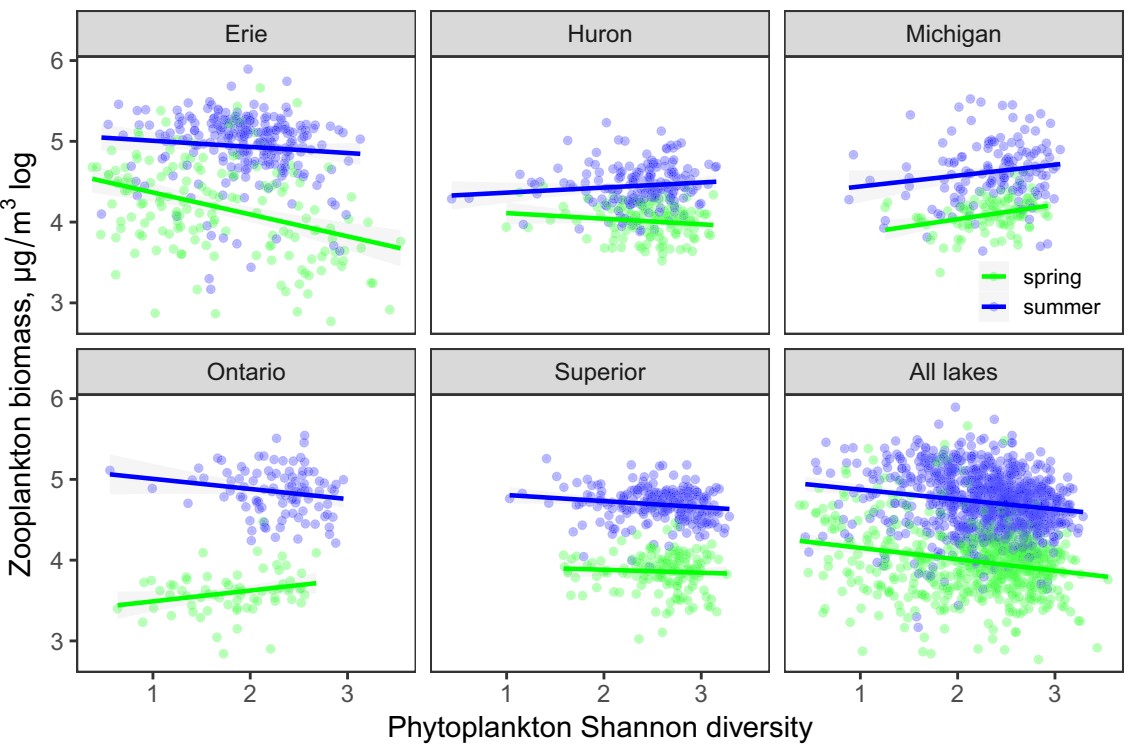

**Fig 3. Total zooplankton biomass as a function of phytoplankton diversity (Shannon H).**

reproduction. The relationships of zooplankton biomass and diversity with edible phytoplankton were similar to those with total phytoplankton biovolume, likely because edible and total phytoplankton biovolume were closely correlated in all lakes with exception of Lake Erie, the most productive lake with a greater incidence of harmful algal blooms. Although other studies have shown that the proportion of inedible phytoplankton, particularly Cyanobacteria, increases in higher productivity lakes [16, 55, 56], cyanobacteria can also be abundant in oligotrophic systems [57] and can constitute a considerable part of the total biomass across large total phosphorus gradients [58]. Increasing biomass of less-edible phytoplankton, such as Cyanobacteria, has been observed to limit zooplankton resource use efficiency and the structure of trophic interactions [16]. However, the relationship between cyanobacterial blooms and zooplankton is variable, and previous studies have observed positive correlations between cyanobacteria concentrations and several groups of zooplankton [19].

Bottom-up forcing was demonstrated to be important in Lakes Michigan and Huron [59], where declines in zooplankton biomass and particularly herbivorous cladocerans were associated with simultaneous declines in spring chlorophyll indicating potential grazer limitation [36, 59, 60]. In other cases, changes in zooplankton are better explained by top-down forcing through increased invertebrate or fish predation [30, 33, 39], including changes in vertical distribution [61]. It is likely that the relative importance of these forces varies across the large spatial and trophic gradient and with season, contributing to the overall uncertainty in the zooplankton-phytoplankton relationship.

Zooplankton biomass was weakly negatively correlated with algal diversity, and it is possible that counteractive effects of algal diversity can be manifested through improved chances of balanced nutrition vs. dilution of the most nutritious taxa [13]. This effect sign was the opposite

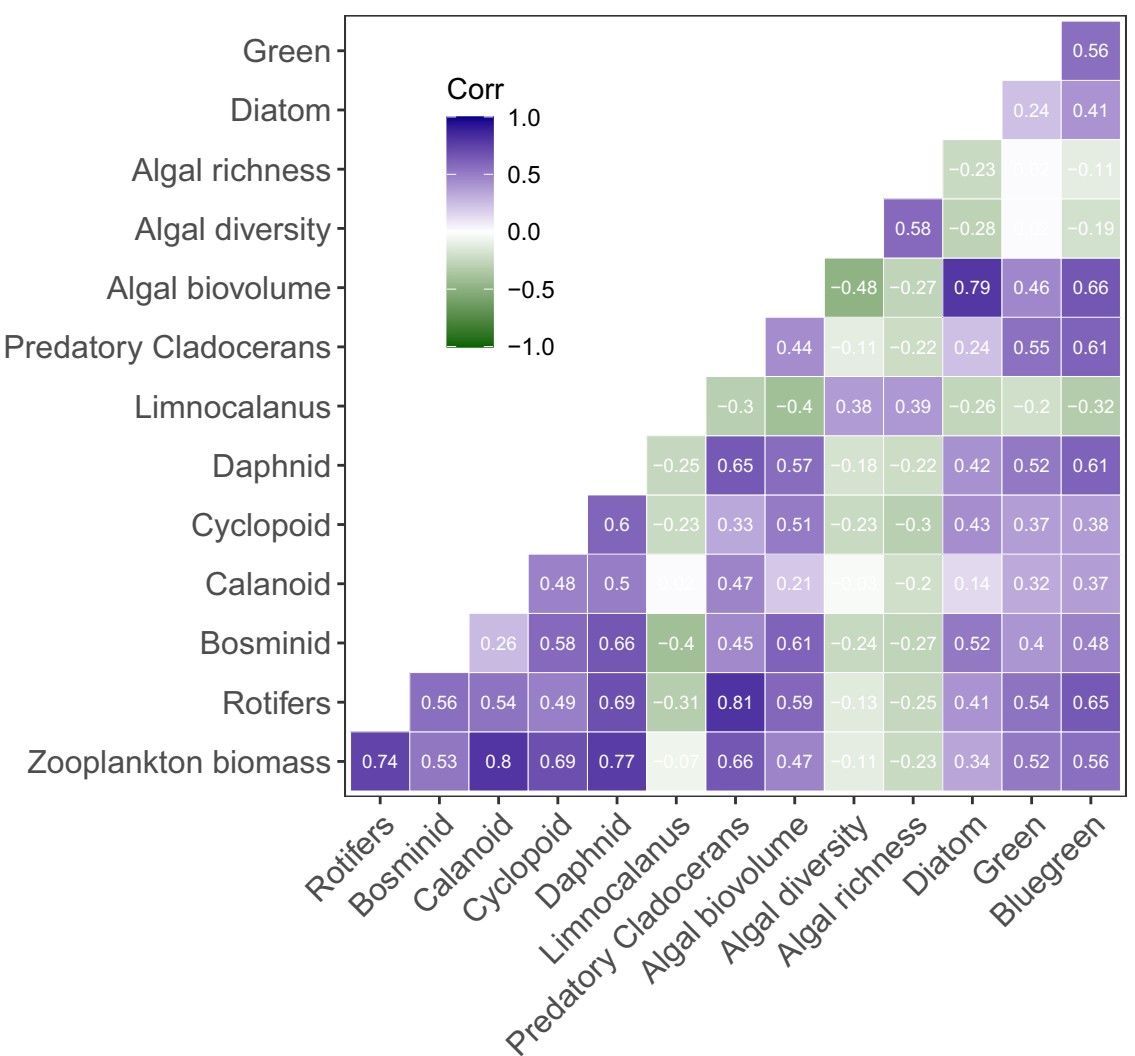

**Fig 4. Cross-correlation matrix for key groups of zooplankton and phytoplankton.** Spearman correlation coefficients color-coded by shade intensity; all biovolume and biomass metrics have been log$_{10}$-transformed. Relationships with visible R have P < 0.0001, whereas relationships with R < 0.10 are displayed as white text on light background.

of the one we expected based on prior studies [12, 13] possibly because pelagic Great Lakes do not include highly eutrophic waters, where extreme cyanobacterial dominance (and therefore decreased overall algal diversity) is more likely to reduce availability and diversity of preferred algal resources to the extent detrimental to consumers. Zooplankton and phytoplankton Shannon diversity were not significantly correlated in our study, providing additional evidence for inconsistent vertical diversity effects across aquatic ecosystems. Positive vertical diversity effects have been observed between bacterial and nanoflaggelate assemblages [62]; however, zooplankton diversity was not predicted by phytoplankton diversity across a wide range of marine systems [63], tropical streams [64], or temperate lakes [65].

We observed stronger correlations between the different zooplankton groups (with a particularly high correlation between predatory cladocerans and rotifers) than between zooplankton

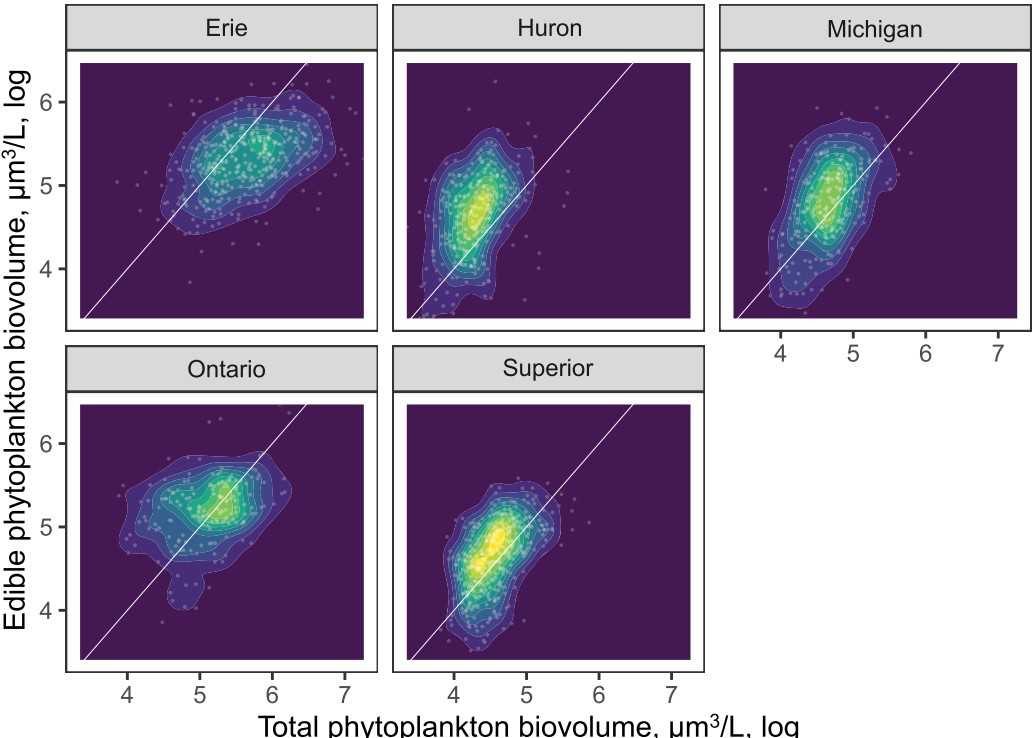

**Fig 5. Relationship between edible and total phytoplankton biovolume across the Great Lakes.** White line indicates the 1:1 ratio, the degree of departure from this line illustrates decreasing relative biovolume of edible algal taxa.

and phytoplankton. This may indirectly suggest a lack of strong feeding selectivity for zooplankton feeding on phytoplankton, at least at the division level, as well as a lack of general avoidance by zooplankton of Cyanobacteria [45], ability to adapt [66], or masking of feeding selectivity by other confounding factors. One of those factors could be availability of picoplankton, which could make an important contribution to the diets of smaller zooplankton. The predator-prey ratio of the zooplankton assemblage was weakly positively predicted by algal diversity, providing marginal support for our hypothesis that more diverse algal assemblages may support greater predator densities, which may not be surprising in the light of the overall weak links between zooplankton and phytoplankton in this system.

It is important to note that over these time scales, our dataset has temporal sampling limitations (only 2 sampling events/station/year) and lower number of stations sampled in the earlier years. Integrated samples are collected from the isothermal upper layer of the water column to favor even sampling of the phytoplankton assemblage. Although we did not see time of sampling explaining additional variation, other studies have shown that many zooplankton species have pronounced vertical migration [67–69] which could further contribute to the observed uncertainty about zooplankton-phytoplankton relationships. All of these factors may limit our ability to draw conclusions about the strength of temporal trends across the entire study period.

Understanding the relationships between phytoplankton and zooplankton is important for predicting the effects of climate change and nutrient loading on food web structure and higher trophic level [54, 59]. A close correspondence between primary producer and consumer assemblages, indicative of bottom-up regulation, can make consumer populations more vulnerable to changing algal phenology and decreased overall lake productivity. However, we did not observe a close correspondence in the Great Lakes, making it more difficult to predict how

the higher trophic levels would be affected by the continued changes in phytoplankton assemblages.

## Supporting information

**S1 Fig. Vertical diversity effects, or correlations between phytoplankton and zooplankton diversity.** Data are presented for all lakes and for individual Laurentian Great Lakes by season.
(EPS)

**S2 Fig. Zooplankton predator-prey (i.e., zooplanktivore-grazer) ratios as a function of attributes of phytoplankton assemblage.** Blue line indicates a Generalized Additive Model (GAM) fit. Algal biovolume is in $\mu m^3$/L, $\log_{10}$ transformed; other metrics are diversity and richness.
(EPS)

**S3 Fig. Edible phytoplankton biovolume and zooplankton biomass correlations by season.**
(EPS)

**S4 Fig. Total zooplankton biomass as a function of only edible phytoplankton diversity.**
(EPS)

**S5 Fig. Herbivorous zooplankton biomass as a function of only edible phytoplankton biovolume.**
(EPS)

**S6 Fig. Herbivorous zooplankton biomass as a function of only edible phytoplankton Shannon diversity.**
(EPS)

**S7 Fig. Effects of sampling time on zooplankton biomass-edible algal biovolume correlations within lakes.**
(EPS)

**S8 Fig. Temporal dynamics of the edible algal biovolume as a function of total algal biovolume.** Data are presented across all Great Lakes.
(EPS)

**S1 Table. Total number of stations sampled by year, lake and season.** Lakes: ER–Erie, HU–Huron, MI–Michigan, ON–Ontario, SU–Superior and seasons: Spr–spring, Sum–summer.
(CSV)

**S2 Table. Summary of phytoplankton data with edibility rankings by shape and nutritional content.** SPECCODE–standard species code; maxRelBiov–maximum relative biovolume within a sample (indicator of relative importance combined with frequency), frequency–number of samples in which the taxon was detected; DIV–division; SPECIES–species name, nutrition edibility and shape edibility–categorical rankings.
(CSV)

**S1 Data.**
(XLSX)

## Author Contributions

**Conceptualization:** Katya E. Kovalenko, Euan D. Reavie, Stephanie Figary, Lars G. Rudstam, James M. Watkins, Anne Scofield, Christopher T. Filstrup.

**Data curation:** Euan D. Reavie, Stephanie Figary, Lars G. Rudstam, James M. Watkins, Anne Scofield.

**Formal analysis:** Katya E. Kovalenko, Stephanie Figary.

**Funding acquisition:** Euan D. Reavie, Lars G. Rudstam.

**Investigation:** Lars G. Rudstam, James M. Watkins, Anne Scofield.

**Methodology:** Euan D. Reavie, Stephanie Figary, Lars G. Rudstam, James M. Watkins, Anne Scofield, Christopher T. Filstrup.

**Project administration:** Euan D. Reavie, Lars G. Rudstam, James M. Watkins, Anne Scofield.

**Resources:** Katya E. Kovalenko, Euan D. Reavie, Stephanie Figary, Lars G. Rudstam, James M. Watkins, Anne Scofield.

**Supervision:** Euan D. Reavie, Lars G. Rudstam, James M. Watkins.

**Validation:** Stephanie Figary.

**Visualization:** Katya E. Kovalenko.

**Writing – original draft:** Katya E. Kovalenko.

**Writing – review & editing:** Katya E. Kovalenko, Euan D. Reavie, Stephanie Figary, Lars G. Rudstam, James M. Watkins, Anne Scofield, Christopher T. Filstrup.

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
