## [Decision Letter · Decision Letter 0]

11 Apr 2023

PONE-D-23-03077Zooplankton-phytoplankton biomass and diversity relationships in the Great LakesPLOS ONE

Dear Dr. Kovalenko,

Thank you for submitting your manuscript to PLOS ONE. After careful consideration, we feel that it has merit but does not fully meet PLOS ONE’s publication criteria as it currently stands. Therefore, we invite you to submit a revised version of the manuscript that addresses the points raised during the review process. Two reviewers see merit in the manuscript, but both suggest major revision. Reviewer 1 emphasizes a lack of detail in the methods, and the need to temper some statements in the light of tenuous correlations.  This reviewer provided a marked up copy of the manuscript with comments and edits. I urge you to look carefully at that document. The second reviewer is concerned that you are missing recent relevant literature and that your discussion misses some mechanistic explanations.  Please, carefully consider both reviewers' comments and suggestions if you decide to submit a revised version of the manuscript. 

We look forward to receiving your revised manuscript.

Kind regards,

Hans G. Dam, Ph. D.

Academic Editor

PLOS ONE

Reviewers' comments:

Reviewer's Responses to Questions

**Comments to the Author**

1. Is the manuscript technically sound, and do the data support the conclusions?

Reviewer #1: Yes

Reviewer #2: Yes

2. Has the statistical analysis been performed appropriately and rigorously? 

Reviewer #1: Yes

Reviewer #2: Yes

3. Have the authors made all data underlying the findings in their manuscript fully available?

Reviewer #1: Yes

Reviewer #2: Yes

4. Is the manuscript presented in an intelligible fashion and written in standard English?

Reviewer #1: Yes

Reviewer #2: Yes

5. Review Comments to the Author

Reviewer #1: • The weak correlations presented here should be treated as such and conclusions based on these relationships should be considered with caution to not over emphasize their significance.

• The paper is well written, but limited in the details included in the methods section. Additional details should be included, particularly in the statistical analysis section as how the analysis was accomplished (Time to process, diversity index, spearman correlation, etc)

• Some limitations of the data were brought up (edibility), however all limitations (integrated samples, net speed and size, day/night, only up to 2 sample points per station per year, minimum of 2 stations per lake) need to be discussed and the implications considered.

• Several figure and table captions contained methods or analysis results not discussed in the body of the paper. The authors should review all captions to ensure that all appropriate content is included in the body of the text.

• Additional comments and edit suggestions made directly in the attached PDF.

Reviewer #2: Dear Author,

your study based on the ratio of predator-to-prey biomass as a key element of trophic structure of lake ecosystems. It is typically investigated from a food chain perspective, ignoring channels of energy transfer that may govern community structure where you study try to approach those processes across space and time.

Line 42-43: I would suggest to add some important recent studies on the predator-prey biomass ratio that have shown that can be stable. In general I would suggest to ass more literature on paper that have been testing this ration and explain what they have been observing.

literature example: Perkins et al. 2022 Consistent predator-prey biomass scaling in

complex food webs

Line 50-53: I think there are studies that show changes of size spectra, as well as biomass across time and space. I would suggest to include those studies because there are closely related with what you are testing in this study.

Line 195-196: How do you explain this? This a pattern that have been observed or not by other studies you should include them and try to explain why you studies confirm on not what other paper have found. You describe the patterns but you haven't tried to explain them.

Line 198-199: I don't think that your have tested the size spectra, maybe you mean something else. Please explain.

Line 208-210:Are all the zooplankton species included in the study herbivores. If not then any you have to separate them cause the carnivore zooplankton is most probably not going to respond on the phytoplankton biovolume changes. Also the presence of carnivore zooplankton may affect the size of the herbivore zooplankton that may in turn be affected by the phytoplankton volume. so it may be a combined affect of those two. This is maybe something to include in the discussion session.

I think in general the discussion session is missing important literature about their field as well as some possible mechanistic explanation about the patterns you are observing. But the pattern you observe are interesting!

6. PLOS authors have the option to publish the peer review history of their article (what does this mean?). If published, this will include your full peer review and any attached files.

Reviewer #1: No

Reviewer #2: No

---

## [Author Response · Author response to Decision Letter 0]

16 Jun 2023

please see the letter of responses (attached) for detailed responses to reviewer comments (it is easier to read in that format)

---

## [Decision Letter · Decision Letter 1]

15 Aug 2023

PONE-D-23-03077R1Zooplankton-phytoplankton biomass and diversity relationships in the Great LakesPLOS ONE

Dear Dr. Kovalenko,

I have now received  feedback from the two original reviewers of your manuscript on the revised version. One reviewer stated directly to me: " The revised manuscript addresses all the comments/suggestions made by the reviewers and I'm very pleased that you have extended your analysis. The present revised manuscript makes an excellent an timing contribution to PLOS ONE." The other review is copied below, and you can see that it contains stylistic suggestions and comments to improve the final version of the paper. Once you make those, i can move to accept the paper. Like the reviewer, I was unable to open the attached figures in the manuscript. Please ensure they are in a format that the journal accepts.

We look forward to receiving your revised manuscript.

Kind regards,

Hans G. Dam, Ph. D.

Academic Editor

PLOS ONE

Journal Requirements:

Reviewers' comments:

Reviewer's Responses to Questions

**Comments to the Author**

1. If the authors have adequately addressed your comments raised in a previous round of review and you feel that this manuscript is now acceptable for publication, you may indicate that here to bypass the “Comments to the Author” section, enter your conflict of interest statement in the “Confidential to Editor” section, and submit your "Accept" recommendation.

Reviewer #1: (No Response)

2. Is the manuscript technically sound, and do the data support the conclusions?

Reviewer #1: Yes

3. Has the statistical analysis been performed appropriately and rigorously? 

Reviewer #1: Yes

4. Have the authors made all data underlying the findings in their manuscript fully available?

Reviewer #1: Yes

5. Is the manuscript presented in an intelligible fashion and written in standard English?

Reviewer #1: Yes

6. Review Comments to the Author

Reviewer #1: Thank you to the authors for their revisions.

Please see my comments on the following minor concerns:

- Line 12 (and lines 73 and 83), EPA should defined at first use. If not in the abstract then in the introduction.

- Lines 48-49, this sentence is awkward. the authors should revise the sentence to avoid the double use of "can" and clarify this statement.

- Lines 200-202 (Figure Caption for Figure 5), the authors should include full lake names in the figures for consistency with the other figures included in the text, (not supplemental), or define the acronyms in the caption.

- Line 267, what do the authors mean by "dilution of this pattern". Perhaps this can be reworded for clarity.

- Figure 1, the units should be properly formatted as superscript in the figure axes.

- Figure 2, the units should be properly formatted as superscript in the figure axes.

- Figure 3, the units should be properly formatted as superscript in the figure axis.

- Figure 4, The colors for this figure, particularly for Limnocalanus, and algal diversity are difficult to read on a computer screen and near impossible to read when printed. I strongly recommend the authors choose a different text color as the white text on white correlation is bad.

- Figure 5, the units should be properly formatted as superscript in the figure axis. Additionally, the acronyms as described above.

- FigS2, I can't open this file. Is this file corrupt?

- FigS3, the units should be properly formatted as superscript in the figure axes.

- FigS4, the units should be properly formatted as superscript in the figure axis.

- FigS5, the units should be properly formatted as superscript in the figure axes.

- FigS6, the units should be properly formatted as superscript in the figure axis.

- FigS7, the units should be properly formatted as superscript in the figure axes. Additionally, 'Spr' and 'Sum' do not need to be shortened here, they could easily be written out without impacting the figure.

7. PLOS authors have the option to publish the peer review history of their article (what does this mean?). If published, this will include your full peer review and any attached files.

Reviewer #1: No

---

## [Author Response · Author response to Decision Letter 1]

29 Sep 2023

Reviewer #1: Thank you to the authors for their revisions.

Please see my comments on the following minor concerns:

- Line 12 (and lines 73 and 83), EPA should defined at first use. If not in the abstract then in the introduction.

______ Defined as suggested.

- Lines 48-49, this sentence is awkward. the authors should revise the sentence to avoid the double use of "can" and clarify this statement.

______ Thank you for pointing this out, rephrased as “Predator-prey biomass ratios can respond to environmental stressors when predators take longer to recover from perturbations, e.g., in isolated environments [20]; however, other studies show remarkable consistency in predator-prey ratios across a wide range of taxa and systems [21,22].”

- Lines 200-202 (Figure Caption for Figure 5), the authors should include full lake names in the figures for consistency with the other figures included in the text, (not supplemental), or define the acronyms in the caption.

_______ Done, full lake names included.

- Line 267, what do the authors mean by "dilution of this pattern". Perhaps this can be reworded for clarity.

______ Clarified as “masking of feeding selectivity by other confounding factors”, meaning that we cannot exclude presence of feeding selectivity however it may be masked by other, stronger drivers of phytoplankton abundance.

- Figure 1, the units should be properly formatted as superscript in the figure axes.

_______ Done.

- Figure 2, the units should be properly formatted as superscript in the figure axes.

_______ Done.

- Figure 3, the units should be properly formatted as superscript in the figure axis.

_______ Done.

- Figure 4, The colors for this figure, particularly for Limnocalanus, and algal diversity are difficult to read on a computer screen and near impossible to read when printed. I strongly recommend the authors choose a different text color as the white text on white correlation is bad.

______In this case, we adjusted the color of the text and background specifically so that non-significant correlations (with very low R values) do not show up. Because this was intentional, we have not changed the figure but clarified it better in the legend.>>>

- Figure 5, the units should be properly formatted as superscript in the figure axis. Additionally, the acronyms as described above.

_______ Done.

- FigS2, I can't open this file. Is this file corrupt?

- FigS3, the units should be properly formatted as superscript in the figure axes.

- FigS4, the units should be properly formatted as superscript in the figure axis.

- FigS5, the units should be properly formatted as superscript in the figure axes.

- FigS6, the units should be properly formatted as superscript in the figure axis.

- FigS7, the units should be properly formatted as superscript in the figure axes. Additionally, 'Spr' and 'Sum' do not need to be shortened here, they could easily be written out without impacting the figure.

_______ All supplemental figures have been redone to address reviewer comments. Fig S2 was redone to ensure it is not corrupt. The current version opens fine, but all .eps figures (as required by the journal) need a postscript reader or need to be converted to a .pdf.

---

## [Editor Report · Decision Letter 2]

4 Oct 2023

Zooplankton-phytoplankton biomass and diversity relationships in the Great Lakes

PONE-D-23-03077R2

Dear Dr. Kovalenko,

We’re pleased to inform you that your manuscript has been judged scientifically suitable for publication and will be formally accepted for publication once it meets all outstanding technical requirements.

Kind regards,

Hans G. Dam, Ph. D.

Academic Editor

PLOS ONE
---

## [Editor Report · Acceptance letter]

19 Oct 2023

PONE-D-23-03077R2 

Zooplankton-phytoplankton biomass and diversity relationships in the Great Lakes 

Dear Dr. Kovalenko:

I'm pleased to inform you that your manuscript has been deemed suitable for publication in PLOS ONE. Congratulations! Your manuscript is now with our production department. 

Kind regards, 

on behalf of

Dr. Hans G. Dam 

Academic Editor

PLOS ONE